# Fast Route Planner Considering Terrain Information

**DOI:** 10.3390/s22124518

**Published:** 2022-06-15

**Authors:** Jonghoek Kim

**Affiliations:** Electronic and Electrical Department, Sungkyunkwan University, Suwon 03063, Korea; jonghoek@gmail.com

**Keywords:** fast route planner, collision avoidance, rough terrain, safe route planner, route length, terrain information

## Abstract

Route planning considering terrain information is useful for the navigation of autonomous ground vehicles (AGV) on complicated terrain surfaces, such as mountains with rivers. For instance, an AGV in mountains cannot cross a river or a valley that is too steep. This article addresses a novel route-planning algorithm that is time-efficient in building a sub-optimal route considering terrain information. In order to construct a route from the start to the end point in a time-efficient manner, we simulate two virtual vehicles that deploy virtual nodes iteratively, such that the connected node network can be formed. The generated node network serves as a topological map for a real AGV, and we construct the shortest route from the start to the end point utilizing the network. The route is weighted considering the route length, the steepness of the route, and the traversibility of the route. Through MATLAB simulations, we demonstrate the effectiveness of the proposed route-planning algorithm by comparing it with RRT-star planners.

## 1. Introduction

Route planning considering terrain information is useful for the navigation of autonomous ground vehicles (AGV) on complex terrain surfaces such as mountains. For instance, an AGV in mountains cannot cross a river or a valley. Moreover, the AGV may have difficulty in maneuvering along a route that is too steep. In this article, we assume that the AGV can plan its route while accessing a contour map, which shows the terrain information of the environments. This article addresses a novel route-planning algorithm that is time-efficient in building a sub-optimal route considering terrain information.

A simple 2D route planner cannot be applied on complex terrain surfaces, where there are many environment constraints and uncertainties [1]. Route planners considering terrain information are required, especially in complicated terrain surfaces such as mountains.

It is desirable that a route plan searches for a safe route in a time-efficient manner. This paper proposes a novel route-planning algorithm that is fast in building a sub-optimal route. Since our route-planning algorithm runs quickly, the proposed planner is useful for the on-line planning of an autonomous robot with cheap embedded systems.

There are many papers on route planning for robots [1,2]. A${}^{*}$ [3], Theta${}^{*}$ [4,5], Ant Colony Optimization (ACO) [6], and Voronoi graph [7] have been applied as route planners of various AGVs. ACO, A${}^{*}$, and Theta${}^{*}$ require that the entire region is already completely covered by multiple grid cells. ACO has advantages such as good feedback information and better distributed computing. However, it has some problems such as slow convergence and prematurity [6]. A${}^{*}$ or Theta${}^{*}$ work to find a minimum-cost route from the start to the end point through graph searching [1]. As one increases the size of the entire region, the number of grid cells covering the entire region also increases. Therefore, calculating a route under ACO, A${}^{*}$, or Theta${}^{*}$ on a large terrain surface may not be viable due to the large computational load.

There are many papers on route planning considering terrain information [8,9,10,11,12]. The authors of [8] presented a control architecture for fast quadruped locomotion over rough terrain. The goal of the high-level planner in [8] was to determine a set of feasible footsteps across the terrain, ideally one that is robust to minor deviations and slips of the robot. The first step in the high-level planning of [8] was to build a “foot cost map” that indicates the desirability of stepping at any given point in the terrain. Using the map, ref. [8] applied a Dynamic Programming (DP) algorithm to plan a minimum-cost path across the terrain. The authors of [9] addressed using different resolution terrain passability maps to construct graphs, which allow for the determination of the optimal route between two points. The routes were generated using two path planners: Dijkstra’s and A${}^{*}$. In [10], a modified approach of the D${}^{*}$ algorithm was proposed. In addition to the distance to be traveled, terrain slope estimate was also used in the computation of the cost function to plan the route. Reference [11] addressed a traversability assessment method and a trajectory planning method considering a rough terrain surface. The planners in [11] find an initial route through the non-holonomic A${}^{*}$ planner. Then, the initial route is optimized in terms of the traversability utilizing Lagrange multipliers.

The route planner in [12] starts offline by computing several potential paths to the end using A${}^{*}$ such that each path can later provide suitable options to the AGV if replanning is required due to unexpected mobility difficulties. The AGV in [12] gains information about its environment as it drives and updates the map locally if major discrepancies are found. If an update is made, the remaining driving time along the different options is recalculated, and the most efficient path is chosen.

However, DP [8], A${}^{*}$ [9,11,12], or D${}^{*}$ [10] require that the entire region is already completely covered by multiple grid cells. As one increases the entire workspace size, the number of grid cells covering the workspace also increases. Therefore, calculating a route under DP, A${}^{*}$, or D${}^{*}$ on a large terrain surface may not be feasible due to large computational load.

Reference [13] addressed traversability analysis and route-planning algorithms for tethered rovers operating on steep terrains. Reference [13] ignored the fact that as the elevation angle of a route increases, the vehicle has more difficulty in traversing the route. In our paper, we set a route weight, which increases proportionally to the elevation angle of the route. In this way, we considered the fact that as the elevation angle of a route increases, the vehicle has more difficulty in traversing the route.

Considering a region without grid cells, RRT planners in [14,15,16] utilized random sampling to build a viable route. In RRT${}^{*}$, random samplings generate a viable route, and the route converges to an optimum one as infinite time is spent. In [17], a route planner was proposed, which considered the rough terrain traversability of a vehicle during the tree expansion of RRT${}^{*}$. If a generated route is too steep, then it may be impossible to make the vehicle traverse along the route. Considering this aspect, Ref. [17] set the maximum elevation angle for a traversable passage. Through MATLAB simulations, we demonstrate the effectiveness of our route-planning algorithm by comparing it with RRT${}^{*}$ in [17].

We address the proposed route planner briefly. In order to construct a route from the start to the end point in a time-efficient manner, we simulate that two virtual vehicles deploy virtual nodes iteratively such that the connected node network can be formed. Our vehicle starts from the start and finds a route to the end point. On the other hand, the other vehicle starts from the end point and finds a route to the start point. The size of every virtual vehicle is set considering the safety margin of the generated route. The node network deployed by two virtual vehicles is utilized as the AGV’s topological map, and we build the shortest route from the start to the end point utilizing the network. Each edge in the route is weighted considering the route length, the steepness of the route, and the traversibility of the route.

As one increases the size of the entire region in ACO [6], DP [8], A${}^{*}$ [9,11,12], D${}^{*}$ [10], or Theta${}^{*}$ [4,5], the number of grid cells covering the entire region also increases. Our paper does not use grid cells to cover the entire region, and the shortest route is built using the node network deployed by two virtual vehicles. The proposed route planning algorithm runs quickly in building a sub-optimal route considering a cluttered region without grid cells. This paper proves that our route planning considering terrain information is ensured to build a sub-optimal route from the start to the end point in finite time. Through MATLAB simulations, we demonstrate the effectiveness of our route-planning algorithm by comparing it with RRT${}^{*}$ planners in [17].

The remainder of this article is organized as follows: Section 2 introduces assumptions and definitions. Section 3 addresses a fast route-planning algorithm considering terrain information. Section 4 provides MATLAB simulation results of our route-planning algorithm. Section 5 provides conclusions.

## 2. Assumptions and Definitions

This article addresses a novel route plan that is time-efficient in building a sub-optimal route considering terrain information. We assume that the AGV can plan its route while accessing a contour map indicating the terrain information of the environments. We solve the following problem: *construct a safe route from the start to the end point so that the AGV can safely reach the end point in a time-efficient manner.*

We construct a route considering the AGV, which is approximated as a circle with radius *r*. We utilize two *virtual vehicles*, $R_{1}$ and $R_{2}$, for planning the route of the AGV. $R_{i}$(i∈{1,2}) is simplified as a circle with radius *r* centered at R${}_{i}\in R^{2}$.

We plan the route so that a virtual vehicle $R_{i}$(i∈{1,2}) with radius *r* does not meet with obstacles as it tracks the generated route. As we increase *r*, we increase the safety margin of the vehicle. This increase improves the AGV’s safety.

In order to construct a route from the start to the end point in a time-efficient manner, we simulate two virtual vehicles that deploy virtual nodes iteratively such that the connected node network can be formed. Our vehicle $R_{1}$ starts from the start and finds a route to the end point. The other vehicle $R_{2}$ starts from the end point and finds a route to the start point. The node network generated by $R_{i}$ (i∈{1,2}) is utilized as a topological map for the AGV, and the AGV builds the shortest route from the start to the end point utilizing the network. Every edge in the route is weighted considering the route length, the steepness of the route, and the traversibility of the route.

Note that $R_{i}$ is utilized to construct a real AGV route under simulations. Since $R_{i}$ moves in simulated (virtual) environments, we can make it move with unbounded speed in simulations. The term “virtual” implies that the vehicle exists in simulated environments. Note that as a real AGV tracks the generated route, it cannot move with infinite speed due to hardware limits.

In practice, an AGV cannot cross obstacles such as rivers. We assume that the AGV can access the obstacle information in environments. We say that $R_{i}$
*meets* with an obstacle if the circle centered at $R_{i}$ meets the interior of the obstacle. We plan the route of $R_{i}$ so that $R_{i}$ with radius *r* does not meet with obstacles while traversing the generated route.

For convenience, let $L(c_{1},c_{2})$ define a straight line segment connecting two points $c_{1}\in R^{2}$ and $c_{2}\in R^{2}$. We say that $L(c_{1},c_{2})$ is *obstacle-free* once the following requirement is met: As $R_{i}$ with radius *r* tracks $L(c_{1},c_{2})$, $R_{i}$ does not meet with obstacles.

In order to construct a route from the start to the end point in a time-efficient manner, $R_{i}$ deploys virtual nodes iteratively, such that the connected node network can be formed. We then build the shortest route from the start to the end point utilizing the node network. The route is weighted considering the route length, the steepness of the route, and traversibility of the route.

Each virtual node has circular sensing coverage with radius $r_{s}$. Let the *footprint* of a node *n* define the circle whose center corresponds to *n* and the radius of which is $r_{s}$. Let $S_{n}$ denote the footprint of *n*. Let $n\in R^{2}$ indicate the 2D location of node *n*.

Recall that $R_{i}$ deploys virtual nodes iteratively, such that the connected node network can be formed. The connectivity graph *I* is defined as a set I=(V(I),E(I)). Here, V(I) denotes the vertex set and E(I) denotes the directed edge set. Every vertex in V(I) represents a deployed virtual node. Note that a virtual node can be deployed by $R_{1}$ or $R_{2}$.

Every directed edge, say $\left\{n_{i},n_{j}\right\}\in E\left(I\right)$, indicates a directed straight route from $n_{i}$ to $n_{j}$. Since an edge is directed, $\left\{n_{i},n_{j}\right\}\neq \left\{n_{j},n_{i}\right\}$. $\left\{n_{i},n_{j}\right\}\in E\left(I\right)$ implies that $L(n_{i},n_{j})$ is an obstacle-free route and that $∥n_{i}-n_{j}∥<r_{c}$ such that $r_{c}>r_{s}$. Here, $n_{i}$ denotes the 2D location of node $n_{i}$. In MATLAB simulations (Section 4), we utilize $r_{c}=3*r_{s}$.

For instance, Figure 1 depicts *I*, which is composed of five vertices $n_{1},n_{2},n_{3},n_{4},n_{5}$. There exists a rectangular obstacle in this figure. In addition, the length of $r_{c}$ is plotted at the bottom of this figure. Each directed edge in *I* is plotted with dashed directed line segments in Figure 1. Each edge length is shorter than $r_{c}$, and every directed edge is an obstacle-free route.

Each directed edge, say e∈E(I), is weighted by $w\left(e\right):e\to Z^{+}$. $\left\{n_{i},n_{j}\right\}\in E\left(I\right)$ is directed from $n_{i}$ to $n_{j}$. The weight of $\left\{n_{i},n_{j}\right\}$ is set as
(1)$$\begin{array}{c} w\left(\left\{n_{i},n_{j}\right\}\right)=∥n_{i}-n_{j}∥+W_{h}*A\left(n_{i},n_{j}\right) \end{array}$$
where
(2)$$\begin{array}{c} A\left(n_{i},n_{j}\right)=\tan^{-1}\frac{h\left(n_{j}\right)-h\left(n_{i}\right)}{∥n_{i}-n_{j}∥}. \end{array}$$

In (Equation 1), h(n) defines the height of n, which is available utilizing contour maps. In (Equation 2), $A(n_{i},n_{j})$ indicates the elevation angle (steepness) of $n_{j}$ with respect to $n_{i}$. As the steepness $A(n_{i},n_{j})$ increases, it gets more difficult to traverse from $n_{i}$ to $n_{j}$.

In (Equation 1), $W_{h}$ is the weight parameter for steepness of the route, compared to the route length $∥n_{i}-n_{j}∥$. $A(n_{i},n_{j})>0$ implies that traversing the edge $\left\{n_{i},n_{j}\right\}$ make the vehicle move up a hill. Furthermore, $A(n_{i},n_{j})<0$ implies that traversing the edge $\left\{n_{i},n_{j}\right\}$ make the vehicle move down a hill. In MATLAB simulations (Section 4), we use $W_{h}=1$ when $A(n_{i},n_{j})>0$. Furthermore, we use $W_{h}=0.5$ when $A(n_{i},n_{j})<0$. In this way, we penalize the case where the vehicle moves up a hill, compared to the case where the vehicle moves down a hill.

If a route is too steep, then it may be impossible to make the vehicle traverse along the route. Considering this aspect, we set the maximum angle for $A(n_{i},n_{j})$. Let thres denote the maximum angle for convenience. If $∥A(n_{i},n_{j})∥>thres$, then we set the associated weight as $w(\left\{n_{i},n_{j}\right\})=\infty$. In this way, the vehicle does not traverse along a route which is too steep. In MATLAB simulations (Section 4), we use thres=60 degrees.

The weight of an edge (Equation 1) is set, considering both the path length and the steepness of the edge. In practice, an edge weight can be set considering the detailed traversability condition as well. For instance, we can set the detailed traversability class as follows: benign, sandy, and rough [12]. As an edge contains rough terrain, the edge has larger weight, compared to the case where the edge contains benign terrain.

Utilizing the definition of E(I), $R_{i}$ does not meet with obstacles as it tracks an edge in E(I). Hence, *I* is an obstacle-free topological map for the vehicle $R_{i}$. By accessing the weight for every edge in *I*, $R_{i}$ can build the shortest route from one node to every other node.

In simulations, $R_{i}$ is virtual; thus, it can traverse along an edge in E(I) infinitely fast. In our paper, $R_{1}$ starts from the start and finds a route to the end. Furthermore, $R_{2}$ starts from the end and finds a route to the starting point. Let $d_{i}\in R^{2}$ define the destination point for $R_{i}$. $d_{1}$ is the end point, and $d_{2}$ is the start point.

$S_{n}$, the footprint of a node *n*, meets $L(n,d_{i})$ at one point. This point is termed the *destination-closest point*, since it is closest to the destination among all points on $S_{n}$. See Figure 2 for an illustration.

An *open boundary* of a node *n* defines the set of points on $S_{n}$, such that every point in this set is outside of every obstacle. A *frontier* of *n* is the subset of an open boundary, such that every point on the frontier is outside the footprint of every other node.

We next introduce how to construct *FrontierPoints* that discretize a frontier with discrete points. We randomly construct a point, called the *randomFrontierPoint*, on the footprint of *n*. We then construct *q* points on the footprint of *n* so that they are evenly spaced on the footprint. As *q* increases, we obtain densely spaced points on a footprint.

The destination-closest point, the randomFrontierPoint, and evenly spaced *q* points are termed the *ViableFrontierPoints* of *n*. Among these q+2 ViableFrontierPoints, we select a set of *FrontierPoints*, f(n) on a frontier satisfying the following requirements:A point in f(n) is outside the footprint of every other node.The straight line segment connecting *n* and a point in f(n) is an obstacle-free route for $R_{i}$.The distance between every point in f(n) and an obstacle boundary is larger than *r*, the radius of $R_{i}$.

In our route-planning algorithm, $R_{i}$ utilizes a FrontierPoint as a “way-point” for reaching its destination. The first requirement implies that a FrontierPoint is associated with the border between a space covered by footprints and a space covered by no footprints. If every node has no FrontierPoint, then the footprints of all nodes cover the entire region. The second requirement indicates that as $R_{i}$ moves from *n* to any point in f(n), $R_{i}$ does not meet with obstacles. The third requirement is applied to avoid the situation where $R_{i}$ meets with obstacles when $R_{i}$ reaches a FrontierPoint. The second and third requirements ensure that a FrontierPoint is outside of every obstacle.

Note that $R_{i}$ utilizes a FrontierPoint as a “way-point” for reaching its destination. Due to the generating of randomFrontierPoints, our approach is based on random sampling. In other words, whenever we construct a new route, it is distinct from a route that has been generated before.

Figure 3 illustrates $R_{i}$, deploying a new virtual node iteratively. In this figure, $R_{i}$ is plotted with a circle. $R_{i}$ moves inside a tunnel in this figure. The tunnel boundaries are shown with red curves, and the trajectory of $R_{i}$ is shown with a yellow curve. The large dots on the trajectory of $R_{i}$ indicate the virtual nodes deployed by $R_{i}$. The footprint of every virtual node is plotted with a dotted circle. FrontierPoints are plotted with large dots on the footprint of the rightmost node.

## 3. Fast Route-Planning Algorithm

The proposed route-planning algorithm is addressed in Algorithm 1. We discuss Algorithm 1 in detail. Algorithms 2 and 3 are sub-algorithms of Algorithm 1.

At the initial phase of Algorithm 1, we construct two virtual vehicles $R_{1}$ and $R_{2}$. Initially, $R_{i}$ (i∈{1,2}) deploys a virtual node at its location. Whenever $R_{i}$ deploys a new virtual node, the node is connected to the network, and we update *I*. Furthermore, the sensing capability of the node is enabled. Recall that every virtual node has footprint radius $r_{s}$.

**Algorithm 1** route-planning algorithm
1:Generate $R_{1}$ at the start location (location of a real AGV);2:Generate $R_{2}$ at the end location;3:
**repeat**
4:   For all i∈{1,2}, $R_{i}$ deploys a new virtual node, say *n*, at its location;5:   Update *I* utilizing the newly deployed node *n*;6:   Enable the sensing capability of *n*.7:   FrontierGenerate(n) in Algorithm 2;8:   Every FrontierPoint within $r_{s}$ distance from *n* is removed;9:   $R_{i}$.MoveToFrontier in Algorithm 3;10:**until** there exists an obstacle-free route from the start to the end;11:Generate *K* shortest routes from the start to the end utilizing *I*;12:Among *K* shortest routes, find a smoothest route;13:The real AGV tracks the found route;


Algorithm 2 is utilized to construct ViableFrontierPoints. Furthermore, Algorithm 3 is utilized to make $R_{i}$ maneuver to a FrontierNode, say $n^{c}$, that is closest to its destination $d_{i}$. Here, we say that a node is a *FrontierNode* if the node has a FrontierPoint. $R_{i}$ maneuvers to a FrontierNode that is closest to $d_{i}$, since we need to build the shortest route to $d_{i}$. Note that $d_{1}\neq d_{2}$.

In Algorithm 3, $n^{c}$ is a FrontierNode, that is closest to $d_{i}$. In Algorithm 3, $f_{R_{i}}$ is a FrontierPoint of $n^{c}$ that will be visited by $R_{i}$ after $R_{i}$ maneuvers to $n^{c}$.

**Algorithm 2** 
FrontierGenerate(n)

1:Generate the point closest to the destination of *n* and set the point closest to the destination as the first ViableFrontierPoint;2:Generate the randomFrontierPoint on the footprint of *n*, and set the randomFrontierPoint as the second ViableFrontierPoint;3:Generate evenly spaced *q* ViableFrontierPoints on $S_{n}$;4:**for** i = 1:1:q + 2 **do**5:   **if** *i*-th ViableFrontierPoint satisfies the requirements for a FrontierPoint **then**6:     Store the ViableFrontierPoint as a FrontierPoint of *n*;7:   **end if**8:
**end for**



**Algorithm 3** $R_{i}$.MoveToFrontier
1:$R_{i}$ finds a FrontierNode, say $n^{c}$, that is closest to $d_{i}$;2:$R_{i}$ maneuvers to $n^{c}$;3:**for** j = 1:1:q + 2 **do**4:   **if** *j*-th ViableFrontierPoint satisfies the requirements for a FrontierPoint **then**5:     $R_{i}$ sets the *j*-th ViableFrontierPoint as $f_{R_{i}}$;6:     Get out of this for loop;7:   **end if**8:
**end for**
9:$R_{i}$ maneuvers to $f_{R_{i}}$;


Since the first ViableFrontierPoint is the point closest to the destination, $R_{i}$ checks the point closest to the destination of $n^{c}$ before checking other ViableFrontierPoints of $n^{c}$. This makes $R_{i}$ head towards its destination if it is possible. This strategy is utilized for reducing the route length to its destination.

Once $R_{i}$ reaches $f_{R_{i}}$, it deploys a new node at $f_{R_{i}}$. Whenever a new node is deployed, we enable the sensing capability of the node. Thereafter, every FrontierPoint within $r_{s}$ distance from the node is removed.

Algorithm 1 iterates until the newly deployed node *n* builds an obstacle-free route from the start to the end. We then find *K* shortest routes from the start to the end. Among *K* shortest routes, we find a smoothest route and set it as the route for the real AGV. A smooth route is desirable considering the traversibility of the AGV.

We discuss how to find a smoothest route among *K* shortest routes. Let route(k), where k∈{1,2,…,K}, denote the *k*-th route in the *K* shortest routes. Let $route\left(k\right)=\left[p_{1}^{k},p_{2}^{k},\ldots ,p_{end}^{k}\right]$ denote the set of nodes along the route route(k). Let $angle(p_{j}^{k})$ denote the angle formed by two vectors $v_{j+1}=p_{j+1}^{k}-p_{j}^{k}$ and $v_{j}=p_{j}^{k}-p_{j-1}^{k}$. Mathematically, we use
(3)$$\begin{array}{c} angle\left(p_{j}^{k}\right)=acos\left(\frac{dot(v_{j+1},v_{j})}{∥v_{j+1}∥∥v_{j}∥}\right). \end{array}$$

Here, $dot(v_{j+1},v_{j})$ denotes the dot product between $v_{j}$ and $v_{j+1}$. Then, the sharpness of route(k) is defined as
(4)$$\begin{array}{c} s\left(route\left(k\right)\right)=max_{j\in \{2,3,\ldots ,end-1\}}\left(angle\left(p_{j}^{k}\right)\right). \end{array}$$

Among all *K* routes, we find a smoothest route using
(5)$$\begin{array}{c} k^{*}=argmin_{k\in \{1,2,\ldots K\}}s\left(route\left(k\right)\right). \end{array}$$

Then, the real AGV tracks the smoothest route $route(k^{*})$.

Algorithm 1 iterates until the newly deployed node *n* builds an obstacle-free route from the start to the end. We then find *K* shortest routes from the start to the end. We find a smoothest route among *K* shortest routes and set it as the route for the real AGV. However, when Algorithm 1 ends, we may have a case where we cannot find *K* routes from the start to the end. Suppose that we can only find L<K routes from the start to the end, when Algorithm 1 ends. In this case, we find a smoothest route among *L* routes by replacing *K* in (Equation 5) by *L*. The smoothest route is set as the route for the real AGV.

The planning phase in [12] computed several potential paths to the end using A${}^{*}$ such that each path can later provide suitable options to the AGV if replanning is required due to unexpected mobility difficulties. Similarly to [12], *K* routes in our paper can provide suitable options to the AGV if replanning is required due to unexpected mobility difficulties. The AGV gains information about its environment as it moves and updates the map locally if major discrepancies are found. If an update is made, then the remaining driving time along the different options (*K* routes) is recalculated, and the most efficient path is chosen.

### Analysis

Algorithm 1 cannot proceed if $R_{i}$ cannot find any FrontierNode. The following theorem proves that if $R_{i}$ cannot find any FrontierNode, then the entire open space is covered by footprints.

**Theorem** **1.**
*If $R_{i}$ cannot find any FrontierNode, then the entire open space is covered by footprints.*


**Proof.** Under the transposition rule, we prove the following statement: if an open space, which has not been covered by footprints, exists, then $R_{i}$ can find a FrontierNode.Suppose that an open space, which has not been covered by footprints, exists. Let *O* indicate this uncovered open space. Utilizing the definition of a frontier, at least one node has a frontier on the boundary of *O*. Therefore, $R_{i}$ can find this FrontierNode. □

Theorem 1 implies that if $R_{i}$ cannot find any FrontierNode, then the entire open space is covered by footprints. Therefore, the route from the start to the end is generated before $R_{i}$ cannot find any FrontierNode. In this way, Algorithm 1 continues until a route from the start to the end is found.

Theorem 2 proves that the real AGV does not meet with obstacles while moving along the route constructed under Algorithm 1. This implies that Algorithm 1 generates a safe route for the real AGV.

**Theorem** **2.**
*As the real AGV tracks the route constructed under Algorithm 1, it does not meet with obstacles.*


**Proof.** The route from the start to the end is constructed utilizing *I*. Under the definition of *I*, every edge, say $\left\{n_{1},n_{2}\right\}\in E\left(I\right)$, indicates that $L(n_{1},n_{2})$ is an obstacle-free route. Since the real AGV tracks *I* until meeting the end, the real AGV does not meet with obstacles during the maneuver. We have proved this theorem. □

## 4. Matlab Simulation

In this section, we demonstrate the effectiveness of our planning algorithm (Algorithm 1) through MATLAB simulations. In Algorithm 1, we find K=10 shortest routes from the start to the end. We utilize q=100. The footprint radius is $r_{s}=2$, and we set $r_{c}=3*r_{s}$. The radius of the AGV is r=0.1 distance units. We use thres=60 degrees as the maximum elevation angle for the AGV. The end location is [5, 5]. The initial location of the AGV is [47, 47].

To demonstrate the performance of Algorithm 1, we compare Algorithm 1 with the RRT${}^{*}$ in [17]. We implemented 20 Monte-Carlo (MC) simulations to demonstrate the superiority of our planning method. Let $L^{t}$ where t∈{1,2,…,20} denote the weighted route length constructed under the *t*-th MC simulation. Recall that the weight of each edge is set using (Equation 1).

For rigorous comparison using MC simulations, we use the following evaluation values. Let meanL denote the mean of $L^{t}$ for all t∈{1,2,…,20}. Furthermore, let minL denote the minimum value of $L^{t}$ for all t∈{1,2,…,20}. Let maxL denote the maximum value of $L^{t}$ for all t∈{1,2,…,20}. Furthermore, let CT denote the computational time (in seconds) to run one MC simulation using MATLAB 2016. Here, CT is used to analyze the running time, which is relevant for real-time applications.

### 4.1. Cluttered Terrain Surfaces (Scenario 1)

We present the MATLAB simulation results of the proposed route-planning algorithm (Algorithm 1). Considering Scenario 1, Figure 4 depicts 3D terrain surfaces considered in our simulations. Figure 5 depicts the route constructed under one MC simulation of Algorithm 1. The route is shown with blue line segments. In Figure 5, a non-convex obstacle, such as a lake, is plotted, and we plot the contour map. Note that the AGV cannot cross an obstacle.

#### Comparison with the RRT${}^{*}$

Considering Scenario 1, we compare the performance of the proposed plan with the RRT${}^{*}$ in [17]. The RRT${}^{*}$ ends when the distance between a sample point and the end location is shorter than $r_{s}$. Two sample points are neighbors if the relative distance between them is shorter than $r_{s}$. The step size (expansion of the tree within one sampling interval) in the RRT${}^{*}$ is set as $r_{s}$. For fair comparison with the proposed route plan, (Equation 1) is used as weights for an edge between two neighbors.

Once a route is constructed, we can further improve the route by smoothing it. The Open Motion Planning Library (OMPL) library in https://ompl.kavrakilab.org/index.html (accessed on 2 June 2022) has various smoothers. For fair comparison with the RRT${}^{*}$, we do not apply any smoothers.

Considering Scenario 1, Figure 6 depicts one route constructed under one MC simulation of the RRT${}^{*}$. The generated route is marked with blue line segments.

The MC simulation results are summarized in Table 1. In this table, [pro] presents the proposed route planner. [rrt] presents the RRT${}^{*}$. Table 1 shows that Algorithm 1 is comparable to the RRT${}^{*}$, considering the weighted route length. Furthermore, Algorithm 1 runs much faster than the RRT${}^{*}$ (see CT in Table 1). This implies that Algorithm 1 is superior to the RRT${}^{*}$, considering both the route length and the computational load.

### 4.2. Cluttered Terrain Surfaces with a Steep End Position (Scenario 2)

In Scenario 2, we use thres=70 degrees as the maximum elevation angle for the AGV. This scenario considers the case where the AGV can traverse a steep hill. The end location is [10, 23], which is a rather steep position. The initial location of the AGV is [47, 47].

We present the MATLAB simulation results of the proposed route-planning algorithm (Algorithm 1). Considering Scenario 2, Figure 7 depicts the route constructed under one MC simulation of Algorithm 1. In Figure 7, a non-convex obstacle is plotted, and we plot the contour map. The generated route is marked with blue line segments.

Recall that the first ViableFrontierPoint is the point closest to the destination in Algorithm 1. Thus, a virtual plot with $R_{i}$ at a node *n* checks the point closest to the destination of *n* before checking other ViableFrontierPoints of *n*. This makes $R_{i}$ head towards its destination if it is possible. This strategy is utilized to reduce the route length to its destination, as plotted in Figure 7.

#### Comparison with the RRT*

Considering Scenario 2, we compare the performance of the proposed route plan with the RRT${}^{*}$ in [17]. Figure 8 depicts the route constructed under one MC simulation of the RRT${}^{*}$. The generated route is shown with blue line segments.

The MC simulation results are summarized in Table 2. In this table, [pro] represents the proposed route planner. [rrt] represents the RRT${}^{*}$. Table 2 shows that Alg. 1 outperforms the RRT${}^{*}$, considering the weighted route length. Note that Algorithm 1 runs much faster than the RRT${}^{*}$ (see CT in Table 2). This implies that Algorithm 1 outperforms the RRT${}^{*}$, considering both the route length and the computational load.

### 4.3. Effect of Varying Parameters

In this subsection, we present the effect of varying parameters in our paper.

#### 4.3.1. Effect of Varying $R_{s}$

Until now, we set the footprint radius as $r_{s}=2$. Considering Scenario 1, we vary the footprint radius $r_{s}$ in Table 3.

The MC simulation results are summarized in Table 3. This table shows that as $r_{s}$ increases in the proposed planner, the weighted route length decreases while decreasing the computational load CT. As $r_{s}$ increases in the RRT${}^{*}$, the computational load CT decreases. Table 3 shows that Algorithm 1 is superior to the RRT${}^{*}$, considering both the route length and the computational load.

#### 4.3.2. Effect of Varying *K*

Until now, the proposed planner uses K=10. Among *K* shortest routes, we find a smoothest route and set it as the route for the real AGV. Similarly to [12], *K* shortest routes can provide suitable options to the AGV if replanning is required due to unexpected mobility difficulties.

Table 4 checks the effect of changing *K*. In Table 4, we set the footprint radius as $r_{s}=2$. Considering Scenario 2, the MC simulation results with varying *K* are summarized in Table 4. Among the *K* shortest routes, we find a smoothest route and set it as the route for the real AGV. However, finding the smoothest route does not necessarily decrease the weighted route length. Thus, varying *K* does not change the weighted route length significantly. Moreover, Table 4 shows that increasing *K* does not necessarily increase the computational load CT.

## 5. Conclusions

Considering the case where terrain information is available, this article addresses a route-planning algorithm based on two virtual vehicles and virtual nodes. We prove that the proposed algorithm results in a safe route from the start to the end point in finite time. Through MATLAB simulations, we demonstrate the effectiveness of our route-planning algorithm by comparing it with the RRT${}^{*}$. The proposed route planner can be used for supporting path planning of humans [9,18].

In the future, we will demonstrate the effectiveness of the proposed route-planning algorithm utilizing experiments with a real AGV. One limitation of the proposed route planner is that the proposed planner provides a high-level planner for an AGV while not considering unexpected situations. Future work includes research on the integration of lower-level motion-planning algorithms and will focus on on-line risk assessment and decision making under unexpected situations.

## Figures and Tables

**Figure 1 sensors-22-04518-f001:**
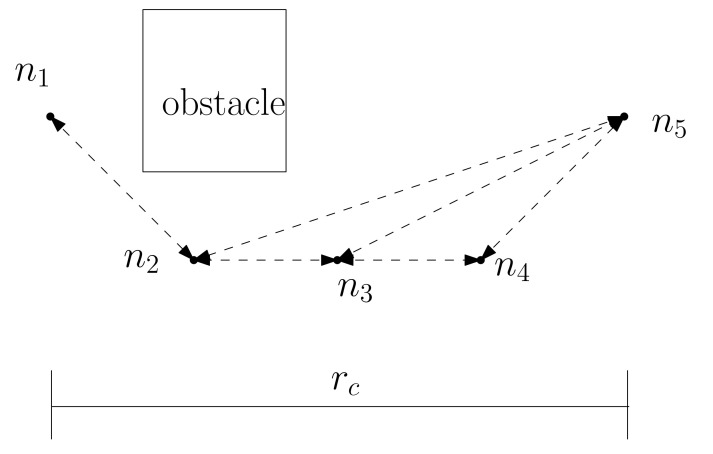
*I* consists of five vertices $n_{1},n_{2},n_{3},n_{4},n_{5}$. There exists a rectangular obstacle. Each directed edge in *I* is plotted with dashed directed line segments. Each edge length is shorter than $r_{c}$, and every directed edge is an obstacle-free route.

**Figure 2 sensors-22-04518-f002:**
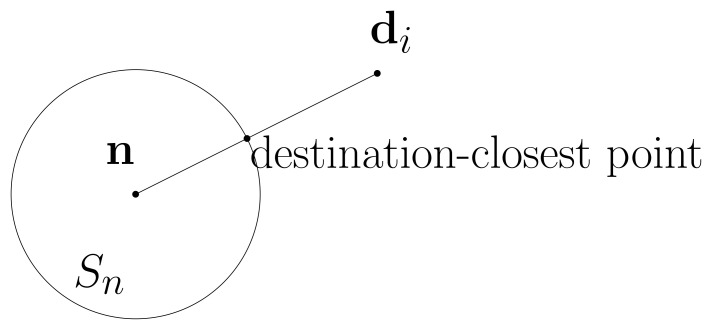
$S_{n}$ meets $L(n,d_{i})$ at one point, termed the destination-closest point.

**Figure 3 sensors-22-04518-f003:**
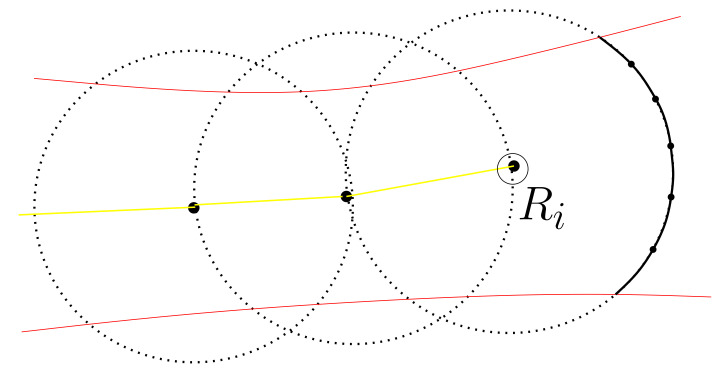
$R_{i}$ keeps deploying a new virtual node. FrontierPoints are plotted with large dots on the footprint of the rightmost node.

**Figure 4 sensors-22-04518-f004:**
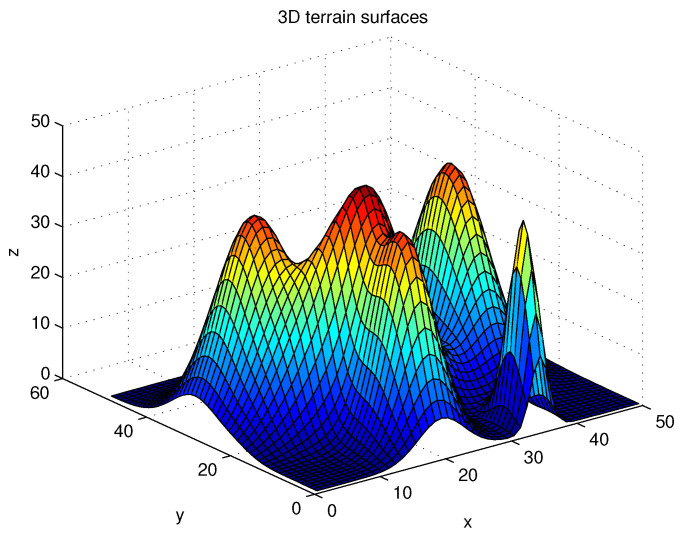
3D terrain surfaces considered in simulations (Scenario 1).

**Figure 5 sensors-22-04518-f005:**
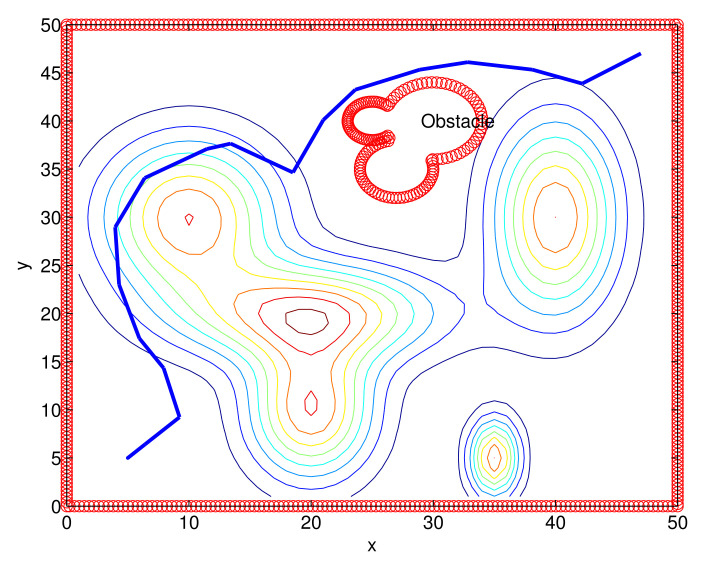
One route constructed under one MC simulation of Algorithm 1 (Scenario 1). A non-convex obstacle is plotted, and we plot the contour map. The route is shown with blue line segments.

**Figure 6 sensors-22-04518-f006:**
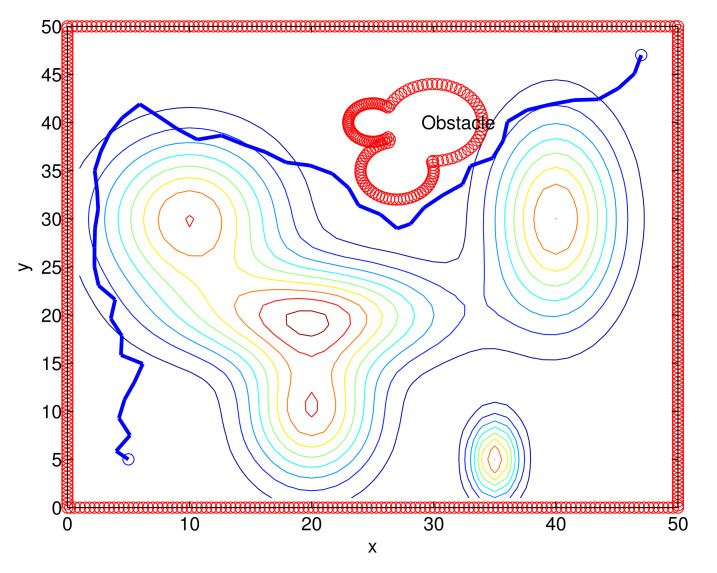
One route constructed under one MC simulation of the RRT${}^{*}$ (Scenario 1). The generated route is shown with blue line segments.

**Figure 7 sensors-22-04518-f007:**
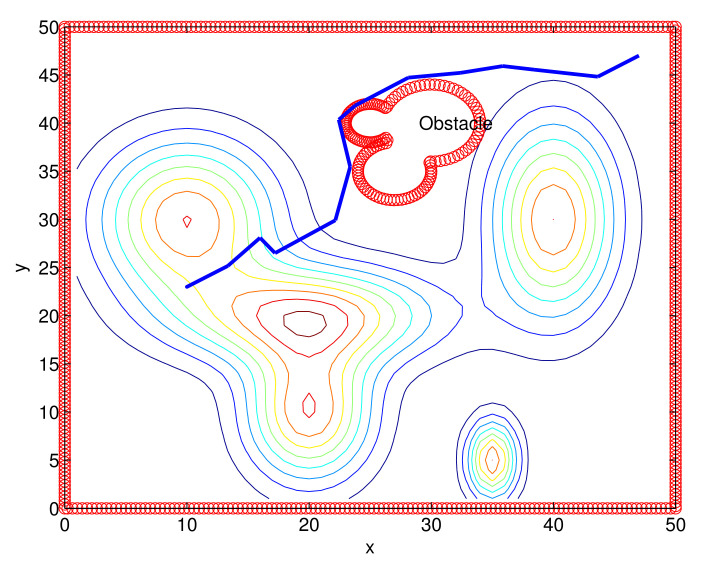
One route constructed under one MC simulation of Algorithm 1 (Scenario 2). A non-convex obstacle is plotted, and we plot the contour map.

**Figure 8 sensors-22-04518-f008:**
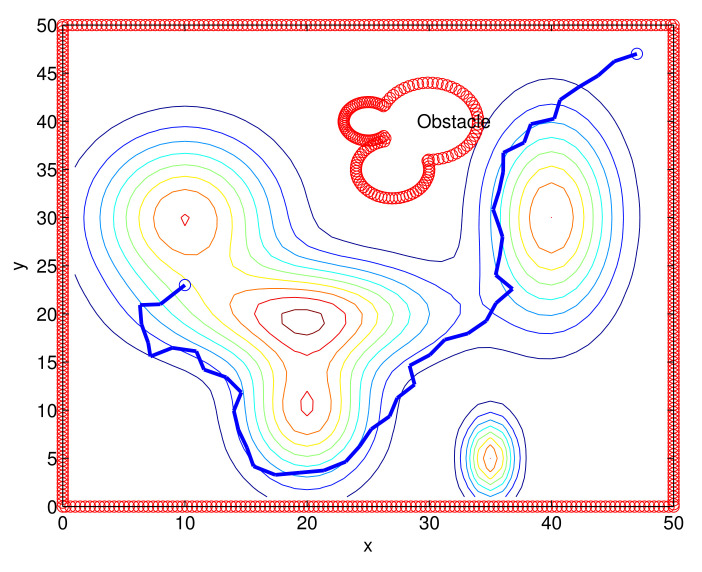
One route constructed under one MC simulation of the RRT${}^{*}$ (Scenario 2). The generated route is marked with blue line segments.

**Table 1 sensors-22-04518-t001:** MATLAB simulation results (scenario 1).

Alg.	MeanL	MinL	MaxL	CT
[pro]	93	73	107	19.5
[rrt]	93	80	110	27.7

**Table 2 sensors-22-04518-t002:** MATLAB simulation results (scenario 2).

Alg.	MeanL	MinL	MaxL	CT
[pro]	50	45	55	3.4
[rrt]	69	53	102	31.3

**Table 3 sensors-22-04518-t003:** MATLAB simulation results with varying $r_{s}$ (scenario 1).

Alg.	$r_{s}$	MeanL	MinL	MaxL	CT
[pro]	2	93	73	107	14.5
[rrt]	2	93	80	110	27.7
[pro]	3	79	72	100	3.2
[rrt]	3	96	77	114	19.3
[pro]	4	78	67	92	1.5
[rrt]	4	93	78	122	17.9

**Table 4 sensors-22-04518-t004:** MATLAB simulation results with varying *K* (scenario 2).

Alg.	*K*	MeanL	MinL	MaxL	CT
[pro]	10	50	45	55	3.4
[pro]	20	49	44	56	2.5
[pro]	30	50	45	58	2.9

## Data Availability

Not applicable.

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
