# Peer review of "Fast Route Planner Considering Terrain Information"

_sensors, 2022, doi:10.3390/s22124518_

Round 1
Reviewer 1 Report
Dear authors,
you have presented a fast route planning algorithm by considering terrain information. Experiments have been performed on MATLAB simulations. The reviewer has the following suggestions and questions.
1. This paper does not contain a related work section. At the end of the introduction section, please consider discussing the relations and differences between your work and existing works.
2. Please consider adding some more efficiency results like the number of parameters, running time, memory requirements, etc. to better illustrate the efficiency of the proposed method. This is very relevant as "fast" is in the title.
3. Please consider discussing some limitations of your work.
4. The proposed method contains many parameters. Please consider conducting some parameter studies to help better understand the method.
5. The results are only performed on MATLAB simulations. Please consider also verifying the proposed method on another existing dataset.
6. Ref. [11] and [12] are the same. One of them can be changed into [*] "Unifying terrain awareness for the visually impaired through real-time semantic segmentation." Sensors 18.5 (2018): 1506.
7. Please consider summarizing some results in a table for comparison against other recent state-of-the-art algorithms, like those published in 2021 and 2022.
8. Would it be possible to conduct a real-world experiment to better verify the effectiveness of the proposed method?
Sincerely,
Author Response
Thank you very much for your valuable comments. The response to Reviewer 1 is attached.

Reviewer 2 Report
The article focuses on fast route planning taking into account terrain information for autonomous ground vehicles and proposes an algorithm to solve this problem. In my opinion, the topic is interesting and quite innovative. This article is well-worded and has good consistency logic. The article presents a series of tests that are intended to enable the reader to understand the method used. In my opinion, after the first in this article, there is very little evidence that the obtained results are better than the results obtained in the article [17], after the second, the size of field data must be comparable in both algorithms, thirdly, the development environment must be the same or comparable etc.
I have a few more comments for users:
1. I suggest adding more references as there are many articles on this topic.
2. I propose a more detailed discussion of the literature review in the introduction.
3. I suggest adding tables to be able to compare the proposed method and the method from article [17].
4. I suggest that the author develops his conclusions more.
Author Response
Thank you very much for your invaluable comments. The response to Reviewer 2 is attached.

Reviewer 3 Report
The manuscript discusses an important branch of autonomous navigation and provides important knowledge for route planning algorithms development.
The manuscript has a logical structure and it's easy to read. The methodology part is well presented and provided algorithms are simulated using the MATLAB software package. The simulation results are clearly presented and seem to be adequate.
I suggest accepting the manuscript to be a part of the MDPI Sensors journal after minor revision. Please provide the next changes before the final submission:
- Manuscript must be formatted according to MDPI Sensors template;
- References [3-8] must be reviewed in detail, citing the works as presented in lines 38 and 39 is not allowed
- Conclusion must be extended
Author Response
Thank you very much for your valuable comments. The response to Reviewer 3 is attached.

Round 2
Reviewer 1 Report
Dear authors,
most of the concerns have been addressed. We suggest that this paper be accepted after a minor revision. For the final version, we would still recommend that the authors analyze the running time and computation complexity results, which are relevant for real-world applications. If your method does not have a clear limitation, please discuss more the trade-offs of your approach.
Sincerely,
Author Response
The response to Reviewer 1 is attached. Thank you very much.

Reviewer 2 Report
I think it is acceptable.
Author Response
Thank you very much for your positive comments.